# Causality-Network-Based Critical Hazard Identification for Railway Accident Prevention: Complex Network-Based Model Development and Comparison

**DOI:** 10.3390/e23070864

**Published:** 2021-07-06

**Authors:** Qian Li, Zhe Zhang, Fei Peng

**Affiliations:** 1Integrated Transportation Research Center, China Academy of Transportation Science, Beijing 100029, China; ecliqian@126.com; 2State Key Laboratory of Rail Traffic Control and Safety, Beijing Jiaotong University, Beijing 100044, China; 3School of Automotive Engineering, Beijing Polytechnic, Beijing 100176, China; pengfei@bpi.edu.cn

**Keywords:** railway accident prevention, critical hazard identification, accident causality network, integer programming

## Abstract

This study investigates a critical hazard identification method for railway accident prevention. A new accident causation network is proposed to model the interaction between hazards and accidents. To realize consistency between the most likely and shortest causation paths in terms of hazards to accidents, a method for measuring the length between adjacent nodes is proposed, and the most-likely causation path problem is first transformed to the shortest causation path problem. To identify critical hazard factors that should be alleviated for accident prevention, a novel critical hazard identification model is proposed based on a controllability analysis of hazards. Five critical hazard identification methods are proposed to select critical hazard nodes in an accident causality network. A comparison of results shows that the combination of an integer programming-based critical hazard identification method and the proposed weighted direction accident causality network considering length has the best performance in terms of accident prevention.

## 1. Introduction

### 1.1. Background

Railway transportation has become the main transportation mode, with the advantages of high speed and low cost. However, railway accidents often interrupt railway transportation processes. Therefore, railway companies emphasize hazard control and emergency management to improve the safety and efficiency of railway transportation. Hazard control is used to alleviate critical or frequent hazard factors and can be considered an accident prevention measure. Emergency management addresses accidents and reduces the negative effects of railway accidents after their occurrence.

This study focuses on identifying critical hazards, which is the core aspect of hazard control. Using accident analysis methods, railway safety managers can investigate the causes of accidents from the aspects of humans, organizations, the environment, and technology. With the increasing number of accident investigation reports, more railway accident causes or hazards can be determined. Typically, some hazards more significantly contribute to accidents than others. These hazards or critical hazards should be identified to support the risk management of railway systems. In the following section, we review the related literature and discuss the contributions of this study.

### 1.2. Related Studies

Experience or data-based risk analysis methods are often used to determine the causes of accidents. The data are typically obtained from accident reports or railway experts. Based on accident data or experience, many accident causation models have been developed to determine hazards that should be controlled or alleviated. Some researchers have classified accident causation models into various groups based on their attention preference toward accidents [1,2,3]. The first group is the domino theory-based accident causation model. The domino model, which was developed by Heintich, considers an accident as a result of several sequential hazard events [4]. Because the domino model simplifies the control of human behavior in accident causation, researchers improved the model by focusing more on management failures, and the modified domino models are defined as management-based accident causation models [5,6]. The third group is human error models, where the occurrence of accidents is attributed to human hazards or errors such as unsafe incorrect responses and improper activities [7]. For example, the human performance railway index operational index has been proposed to predict the probability of human failure in railway operations. Human error analysis, human error reduction, and human factor analysis and classification system (HFACS) methods have been developed to analyze human errors in railway operations [8,9,10,11]. A Poisson regression method was used to determine the relationship between driver personality and driving safety [12]. Many factors affect the safe operation of railway systems in addition to operators, including train drivers, signalers, and controllers. Therefore, the systemic accident models proposed by Hollnagel may be more suitable for railway accident causation analysis [13,14]. An accident causation model for the railway industry has been proposed to investigate the contributions of human failure, technical failure, and external intrusion to final accidents [15]. A system theoretic accident model and process-based model has been proposed to investigate the various causes of railway accidents, including human and management failures [16,17]. Furthermore, the HFACS-RA (Human Factors Analysis and Classification System-railway Accidents) method was proposed to identify and analyze human and organizational factors that affect railway accidents [18]. An integrated evolutionary model called the scenario-risk-accident chain ontology (SRAC) is proposed to determine the risk relevance of railway systems from accident reports [19]. To support the safety management of railway systems, a quantitative causal analysis was proposed to identify the most important factors that contribute to the risk of passenger train accidents [20]. The associated rules have been derived by mining train accident data [21].

Various equipment, machines, and human resources are interrelated in railway systems, which increase the complexity of railway accidents [22,23]. Therefore, network-based accident causation models have been developed to analyze the causes of accidents and model interdependent hazards in recent years [24]. The nodes in the network represent hazard events or accidents, and the links indicate the relationship between hazards and accidents. A directed network has been formulated to analyze the causes of accidents or railway operational accidents [25,26]. The increasing number of accident reports has enabled the weight between hazards to be measured; hence, a directed weighted accident causation network has been proposed to investigate the accident causation complexity [27]. Three hazard control strategies have been compared using the tailored accident causality network (ACN) [28]; however, the hazard control model has not been formulated to identify the optimal hazards to be removed. Some entropy-based methods have been proposed to determine the critical nodes in real-world complex networks [29,30,31] such as power networks [32], biological networks [33], and transportation networks [34,35]; however, there is no method to identify critical hazard nodes in ACNs from the perspective of both global optimization and accident prevention. Therefore, a hazard identification model was developed in this study to select optimal hazards to be removed for railway accident prevention.

The selected critical hazards should block the path from hazards to accidents or lengthen the distance from hazards to accidents to prevent accidents. However, the length of the edges cannot be appropriately obtained; hence, the distance from hazards to accidents cannot be measured, although the proposed ACN model can facilitate the understanding of railway accidents. Herein, we propose a novel accident causation network model inspired by a Bayesian network model [36]; the model enables us to easily measure the weights and lengths of the links. Consequently, the shortest path from hazards to accidents can be obtained, and a critical hazard identification (CHI) model can be formulated. The best CHI model was selected by comparing the performances of different models. The proposed integer programming method performs the best because it can solve the CHI problem from a global perspective.

### 1.3. Contributions and Organization

The proposed model contributes to research pertaining to complex network-based accident analysis methods in three aspects:(1)We propose an improved ACN model (WLDACN) to model the relationship between hazards and accidents. The shortest distances from hazards to accidents can be obtained based on the proposed new length metrics of edges in WLDACN, which is proven to be superior to other previous methods.(2)Given the proposed length metrics of WLDACN, the network efficiency is used to represent the difficulty of hazards causing accidents. Therefore, the accident prevention problem is transferred to successfully minimize the WLDACN efficiency.(3)To support the hazard management of railway systems, we propose a high centrality adaptive and integer programming method to identify critical hazards that greatly contribute to railway accidents. A heuristic algorithm is proposed to solve the integer programming model. The comparison results show that the integer programming method can help prevent accidents better than other models.

The remainder of this paper is organized as follows. Section 2 introduces the ACN construction method and analyzes the objective of hazard control and the formulation of five CHI models. Section 3 presents a real-world case study to verify the effectiveness of the proposed ACN and CHI models. Finally, the conclusions and directions for future research are presented in Section 4.

## 2. Problem Description and Formulation

### 2.1. ACN Model

The first step in our study was to construct a causality network from accident investigation reports, which contain reports of many events. We focus on the description of the accident process because it contains the immediate cause and causal factors of accidents. The causal factors of accidents are typically defined as hazards. Several hazards can be extracted, and the causal relationship between hazards can be described using various causality connectors [37]. For example, we can identify two textual causality connectors such as “because” and “due to” in the sentence “the incident occurred because the driver of the first tram did not stop at the platform or stop signal due to a temporary loss of awareness.” Therefore, the hazard event “loss of awareness” causes the hazard event “tram did not stop at the platform.” We can obtain one hazard pair $\langle i,j\rangle$ if hazard event i causes hazard event j. If hazards i and j are denoted by two nodes, then an arrow from hazard node i to j can be added to represent their causal relationship. After manually extracting all hazard events and accidents from the set of accident investigation reports, we can construct an ACN for railway systems. Each node in a network corresponds to a hazard event or accident. Each edge is associated with a cause–effect pair and directed from the cause (hazard event) to the effect (hazard or accident). Table 1 defines the notations in this paper to formulate the proposed model.

#### 2.1.1. Edge Weight Metrics

We can obtain the same hazard pairs in different accident reports. Therefore, the ACN is a weighted direct accident causality network (WDACN), as shown in Figure 1a. Let S denote the set of nodes in the WDACN. In the WDACN, the weight $w\left(i,j\right)\;$of the edge from node i to node j is equal to the number of hazard pairs $\langle i,j\rangle$ in the reports$\;n_{ij},\;i.e.,:$(1)$$w\left(i,j\right)\;=n_{ij}$$

The causation routes are defined as the set of links from the source hazard node to the accident node. There may be more than one route from source hazard node h to accident node a, and the same route may appear several times. The frequencies of the causation route k from hazard node h to accident node a $F_{k}\left(h,a\right)$ can be defined as $F_{k}\left(h,a\right)\;=\min\left(w\left(i,j\right)\right),i,j\in S_{k}^{h,a}$, where $S_{k}^{h,a}$ is the set of points on causation route k.

As shown in Figure 1a, the causation route (1→2→5) has a lower frequency than the causation route (1→2→4→5). Therefore, the causation path $\left(1\to 2\to 5\right)$ has a lower active probability than the causation path (1→2→4→5). In this study, the causation route with the highest active probability is defined as the most likely causation path (MPCP).

The shortest causation path (SCP) is often used to represent the MPCP from one node to another in a complex network. To analyze the interaction between hazards from the perspective of complex networks, the WDACN was simplified as a direct accident causation network (DACN) [28], as shown in Figure 1b. However, the frequency difference of the hazards cannot be observed in the DACN. As shown in Figure 1c, the MPCP from a to e is $\left(1\to 2\to 5\right)$, which is contrary to the evidence from the WDACN; hence, this method cannot be used to model the interaction between hazard factors and accidents.

Another method is to preserve the frequency information by reversing the weight of the edge [27], as shown in Figure 1c. However, the SCP still cannot indicate the MPCP precisely because paths $\left(1\to 2\to 5\right)$ and $\left(1\to 2\to 4\to 5\right)$ have the same distance (i.e., 1/3), as shown in Figure 1c. Therefore, the SCP resulting from these two methods is not consistent with the MPCP observed from the WDACN.

#### 2.1.2. Edge Length Metrics

To realize the consistency between SCP and MPCP, we propose a new method to measure the length of the edges. First, the weight of the edge $\left(i,j\right)$ in the WDACN can be normalized as follows:(2)$$p\left(i,j\right)=\frac{w\left(i,j\right)}{D_{i}^{out}},$$
where $D_{i}^{out\;}$ is the out-degree of node i. As shown in Figure 1d, the normalized weight $p\left(i,j\right)$ can be interpreted as the active probability of hazard node j for active node i. Let H and A denote the sets of hazard accident nodes. Let $S_{k}^{h,a}$ denote the set of points on causation route k from hazard node h to accident node a. Therefore, the active probability of causation route k can be expressed as:(3)$$P_{k}\left(h,a\right)=\underset{i,j\in S_{k}^{h,a}}{\prod }p\left(i,j\right)$$

$P_{k}\left(h,a\right)$ reflects the conditional probability of causation path k given the occurrence of source hazard h. The MPCP from hazard node h to accident node a can be obtained by solving the following equation:(4)$$MPCP\left(h,a\right)\;=\mathrm{argmax}\left(P_{k}\left(h,a\right)\right),k=1,2,3\ldots K$$

In order to transform the MPCP problem into an SCP problem, the transformation function should satisfy two conditions: (1) it is a monotone decreasing function of probability $p\left(i,j\right)$; (2) the uncertainty of route k can be represented as the sum of uncertainties of each edge. Therefore, logarithmic function is selected as our transformation function. The natural logarithm of Equation (3) can be used as follows:(5)$$-\ln P_{k}\left(h,a\right)=-\sum_{i,j\in S_{k}^{h,a}}\ln(p\left(i,j\right))$$

Equation (5) can denote the distance of route k which is the sum of length of edges. Therefore, the length $l\left(i,j\right)$ of edge $\left(i,j\right)$ can be described as follows:(6)$$l\left(i,j\right)=-\ln\left(p\left(i,j\right)\right)$$

Figure 1e shows the length of edges. The MPCP problem can be transformed into an SCP problem. The length of causation path r from hazard node h to accident node a can be expressed as:(7)$$L_{k}\left(h,a\right)\;=-\ln P_{k}\left(h,a\right)=l_{k}\left(h,a\right),$$
where $l_{k}\left(h,a\right)\;=\sum_{i,j\in S_{k}^{h,a}}l\left(i,j\right)$. Therefore, the SCP can be obtained by solving the following problem:(8)$$SCP\left(h,a\right)\;=\mathrm{argmin}\left(l_{k}\left(h,a\right)\right),k=1,2,3\ldots K$$

The SCP can be obtained using Dijkstra algorithm and used to measure the difficulty of a hazard factor in causing an accident. The probability of hazard node h causing accident a is greater if $SCP\left(h,a\right)$ is shorter. Finally, edges $\left(i,j\right)$ in the ACN exhibit three attributes: weight $w\left(i,j\right)$, active probability$\;p\left(i,j\right)$, and length $l\left(i,j\right)$. Because the proposed ACN contains weights and length metrics and is a directed network, the proposed network model can be named the WLDACN.

### 2.2. CHI Method Development

#### 2.2.1. Objective of CHI

The objective of CHI is to prevent accidents by reducing or eliminating hazards. The ACN efficiency is used as an indicator to reflect the difficulty of hazards causing accidents.
(9)$$E=\underset{h\in H,a\in A}{\sum }\frac{1}{SCP\left(h,a\right)}$$

A lower network efficiency corresponds to a longer SCP from hazards to accidents. Therefore, we should eliminate hazard–accident interactions by removing the nodes in the ACN. We assume that all hazards can be controlled or removed via technological development and management improvement. For example, equipment failures can be eliminated by improving the maintenance strategy, human factors can be controlled via safety training, and environmental hazards can be prevented by monitoring the operation surroundings. Additionally, the cost increases with the number of removed hazard nodes. Therefore, the maximum number of removed hazard nodes in the network should be defined.

Some hazard factors such as wind, snow, and rain cannot be controlled because they originate from the natural environment instead of the railway system. Therefore, we should discuss the controllability of hazard nodes. Hazard factors can be controlled if their causes can be determined, so only hazard nodes with parent nodes can be alleviated.

#### 2.2.2. High Centrality Adaptive Methods

The high centrality adaptive method is typically used to reduce the network efficiency by removing hazard nodes that have the highest centrality. Four methods can be used to measure the centrality of nodes in a complex network: based on the node degree, node betweenness, node closeness, and PageRank [38]. Therefore, high degree adaptive (HDA), high betweenness adaptive (HBA), high closeness adaptive (HCA), and high pagerank adaptive (HPA) methods are used to remove critical hazard nodes in a WLDACN.

The HDA method removes the node with the highest degree centrality (DC) in each iteration. The HDA method recomputes the DCs of the remaining nodes after node removal. The DC of hazard node h can be expressed as:(10)$$D_{h}=D_{h}^{in}+D_{h}^{out}=\underset{i\neq h}{\sum }w\left(i,h\right)\;+\underset{j\neq h}{\sum }w\left(h,j\right)$$

Based on Equation (10), a greater DC of hazard node h results in more causal relationships between hazard node h and other nodes. If a node with a higher DC is eliminated, then more causal relationships can be eliminated.

HBA sequentially removes the node with the highest betweenness centrality (BC) and recomputes the BC for the remaining nodes in each iteration. The BC of hazard node h can be expressed as:(11)$$B_{h}=\underset{\begin{array}{c} i\in H,j\in A\\ i\neq h,j\neq h \end{array}}{\sum }\frac{\rho_{ij}\left(h\right)}{\rho_{ij}},$$
where $\rho_{ij}$ is the number of shortest paths from hazard node i to accident node j, and $\rho_{ij}\left(h\right)$ is the number of shortest paths from hazard node i to accident node j that pass through hazard node h.

Based on the definition of BC, a greater value of the BC of hazard node h results in more SCPs from hazards to accidents to pass hazard h. If a node with a higher BC is eliminated, then more SCPs from hazards to accidents can be removed.

The HCA method removes the node with the highest closeness centrality (CC) and updates the CC for the remaining nodes in each iteration. CC describes the proximity of a hazard node to all other nodes in the ACN. It is calculated as the reciprocal of the average distances from one hazard node to all accident nodes as follows:(12)$$C_{h}=\frac{N_{A}}{\sum_{a\in A}SCP\left(h,a\right)},$$
where $N_{A}$ is the number of accident nodes. Based on Equation (12), a greater $C_{h}$ results in fewer steps from hazard node h to accidents. If the node with the higher CC is eliminated, then the average SCP increases from hazards to accidents.

The HPA method deletes the node with the highest PageRank centrality and subsequently recomputes PageRank for the remaining nodes in each iteration. The PageRank of hazard node h can be expressed as:(13)$$R_{h}=\underset{j\neq h}{\sum }p\left(j,h\right)p\left(j\right)$$

Based on Equation (13), a greater value of $R_{h}$ results in a greater possibility of hazard node h being activated by other hazards. If the nodes with higher PageRank values are eliminated, then the propagation probability from hazards to accidents decreases.

In fact, the hazard link, including hazard h in the source dataset, should be changed if hazard h is deleted. Assuming that we extract one hazard link $\langle i,j,h,a\rangle$ from one accident report, if hazard h has been removed by our CHI methods, then accident a cannot occur; hence, the hazard link should be deleted.

#### 2.2.3. Integer Programming Method

High centrality adaptive methods delete the node with the highest centrality at each iteration, which is a local optimal solution. Therefore, a model should be developed to solve the global optimal solution. From an economical and safety perspective, the CHI problem can be described as follows: To identify the nodes to be removed to reduce the network efficiency, an integer programming model for the CHI problem is formulated, i.e.:(14)$$Min\;\;E\;\left\{\begin{array}{c} \underset{i}{\sum }x_{i}\leq M,\forall i\;\;\;\;\;\;\;\;\;\left(a\right)\\ x_{i}\in \left\{0,1\right\},\forall i\;\;\;\;\;\;\;\;\;\;\left(b\right), \end{array}\right.$$
where $x_{i}$ is a decision variable for removing hazard node i. $x_{i}=1$ indicates that hazard node i is removed. Equation (14a) states that a maximum of M nodes can be removed. Because the objective function of the CHI model cannot be transformed into a linear form, a heuristic algorithm is proposed to solve the integer programming model. The procedures of the algorithm can be described as follows:

Step 1: Based on the controllability of hazards, randomly select M controllable hazard nodes to be deleted from the accident causation network.

Step 2: For each hazard node in a circular order, maintain the other M−1 hazard nodes and identify the optimal solution for the selected hazard node. The solution for the selected hazard node is updated before determining the optimal hazard node in the WLDACN to minimize the network efficiency.

Step 3: Terminate the algorithm when M consecutive solutions for a hazard node do not reduce the network efficiency; otherwise, return to Step 2.

### 2.3. Model Performance Comparison

The objective of the CHI is to increase the overall average degree of difficulty in causing accidents. To compare the performances of CHI methods from a local perspective, the average SCP (ASCP) from all hazards to each accident type and the ASCP from each hazard type to accidents ($ASCP\left(H_{t}\right)$) were used as indicators to evaluate the proposed CHI methods. The ASCP from all hazards to accident type a, $ASCP\left(a\right)$, can be formulated as follows:(15)$$ASCP\left(a\right)=\frac{\sum_{h\in H}SCP\left(h,a\right)}{N_{H}},$$
where $N_{H}$ is the number of hazard nodes. If $ASCP\left(a\right)$ decreases, then the overall probability of all hazards causing accident a increases; otherwise, the overall probability of all hazards causing accident a will decrease.

Hazards can be classified into different types. Each hazard type is composed of various hazard factors. The ASCP from hazard type $H_{t}$ to accident a can be formulated as follows:(16)$$ASCP\left(H_{t},a\right)=\frac{\sum_{h\in H_{t}}SCP\left(h,a\right)}{N_{H_{t}}},$$
where $N_{H_{t}}$ is the number of hazard nodes that belong to hazard type $H_{t}$. Similarly, the ASCP from hazard type $H_{t}$ to all accidents $ASCP\left(H_{t}\right)$ can be formulated as:(17)$$ASCP\left(H_{t}\right)=\frac{\sum_{a\in A}ASCP\left(H_{t},a\right)}{N_{A}}$$

If $ASCP\left(H_{t}\right)\;$decreases, then the overall probability of hazard type $H_{t}$ causing accidents increases; otherwise, the overall probability of hazard type $H_{t}$ causing accidents decreases.

## 3. Case Studies

The proposed model was applied to the hazard management of railway systems. All computational experiments were conducted on a PC with a 2.8-GHz CPU operating the Windows 10 operating system.

### 3.1. Data Description

The data in this study were obtained from RAIB (https://www.gov.uk/government/publications/raib-investigation-reports-and-bulletins (accessed on 1 September 2020)). We obtained 240 accident investigation reports from 2012 to 2020. The hazards were classified into four types: human (H), equipment and machine (EM), environment (E), and management (M). We extracted 20 H-type hazards (H01-H2O), 53 EM-type hazards (EM01–EM53), 12 E-type hazards (E01–E12), and six M-type hazards (M01–M06). Eighteen types of accidents were obtained, as graphically illustrated in Figure 2.

### 3.2. ACN Construction and Analysis

Figure 3 and Figure 4 show the WLDACN based on the interaction between hazard factors and accidents. The weight was computed using Equation (1) and depicted near the edges in the network, as shown in Figure 3. The length of each edge was computed using Equation (6) and depicted near the edges of the network, as shown in Figure 4.

The SCP is an important feature of the WLDACN because it reflects the overall probability of hazards that cause accidents. Table 2 shows the distance from four types of hazards to 18 types of accidents. As shown in Table 2, the derailment accident (A04) had the shortest ASCP (4.44), which indicates that the most likely accident that results from hazards is derailment. Among the four types of hazards causing derailment, the H-type hazard had the shortest path length, which indicates that human failure can much more easily cause derailment. The H-type hazard can much more easily cause 15 types of accidents than the other types of hazards. The EM- and E-type hazards can cause derailment (A04), as indicated by the distance values of 3.54 and 5.62, respectively. M-type hazards were more likely to cause accidents “struck-by (A06),” due to their distance value of 4.78.

Figure 5 shows the ASCP for each hazard type for accidents. As shown in Figure 5, EM-type hazards had the shortest distance to accidents (5.69) among the four types of hazards, which indicates that the most likely cause of accidents is equipment or machine failure. The second most likely cause of accidents is human-type hazards.

Figure 6 shows the ASCP from each H-type hazard node and E-type hazard node to accidents. As shown in Figure 6, the distances from different hazard nodes to accidents were different. H06 (rail line inspector did not identify problems in a timely manner) and H08 (track worker negligence) had shorter ASCPs to accidents than the other H-type hazard nodes. Therefore, hazards H06 and H08 can more easily cause railway accidents than the other H-type hazard nodes. E02 (water hazard) had a shorter ASCP to accidents than other E-type hazard nodes. Therefore, hazard E02 can more easily cause railway accidents than the other E-type hazard nodes.

Railway safety managers should delete as many hazard nodes as possible through hazard control efforts. However, different ASCPs from each hazard node to accidents indicate different contributions of various hazards to railway accidents. Therefore, the CHI model should be used to obtain the critical hazard nodes.

### 3.3. CHI Model Application and Comparison

Based on the obtained WLDACN and hazard controllability analyses, we obtained 64 controllable hazard nodes. Herein, five CHI methods have been proposed to find the critical hazard nodes. Ten nodes were applied to compare these CHI methods, including the HBA, HCA, HDA, HPA, and IPM. The comparison results are shown in Figure 7. Among the four high centrality adaptive strategies, the HBA strategy is much more effective in preventing accidents than the other three methods because it enables more SCPs from hazards to accidents to be removed. It is difficult to distinguish the effectiveness of the HDA and HPA methods because the network efficiency depends on the number of removed nodes. The HCA strategy performed the worst among the four strategies. Therefore, we suggest using the HBA method to identify critical hazards in railway systems among high centrality adaptive strategies. However, the proposed IPM performed better than all high centrality adaptive strategies, as shown in Figure 7. The high centrality adaptive method iteratively removes the nodes with the highest centrality; therefore, it only solves one critical hazard node at each step. However, the IPM obtains the critical hazard nodes from a global perspective. Consequently, the proposed IPM more effectively performed hazard management and accident prevention.

Additionally, the ASCP was used to compare the performances of five CHI methods. Figure 8 shows the ASCP from all hazards for each accident type after removing 10 hazard nodes. We assume that the ASCP is 20 if accidents cannot be caused by hazards. As shown in Figure 8, some accidents cannot be caused by hazards after we remove 10 hazard nodes. Nine types of accidents, including A05, A12, A13, and A14, cannot be caused by hazards when the HBA method is adopted; eight types cannot be caused by hazards when the HCA method is used; nine types cannot be caused by hazards when the HDA method is used; eight types of accidents, including A05, A08, and A14, cannot be caused by hazards when the HPA method is adopted; nine types of accidents, including A05, A08-A14, A16, and A18, cannot be caused by hazards when the IPM method is adopted. Therefore, HBA and IPM better prevent accidents than the other three methods.

Based on Equation (17), Figure 9 shows the ASCP from each hazard type to accidents after we removed 10 hazard nodes. As shown in Figure 9, the IPM and HBA methods performed better than the other three methods because IPM and HBA increased the ASCP of EM-, E-, and M-type more than the HPA, HCA, and HDA methods. The ASCP of H-type hazards can be lengthened by the HDA method to a higher level. However, HAD has a worse overall performance than the IPM and HBA methods, which implies that the proposed IPM and HBA methods significantly reduce the accident-causing probability of each hazard type. Therefore, the IPM and HBA methods can be selected as CHI models to identify the critical hazard nodes. The IPM can reduce the probability of H-, EM-, and E-type hazard-related accidents to a lower level than the HBA. The HBA method can reduce the probability of M-type hazard-related accidents to a lower level than the IPM method. However, the overall length of the ASCP resulting from the IPM was longer than that from the HBA method. Therefore, the proposed IPM is suggested as the CHI method for accident prevention in railway systems.

The hazard nodes to be alleviated were H02 (driver emergency brake failure), H04 (driver’s operation mistake), H06 (rail line inspector did not identify problems in a timely manner), H12 (level crossing watchman’s mistake), H13 (train maintainer’s inadequate maintenance), EM02 (damaged track), EM04 (false signal displayed), EM24 (equipment failure signal), EM29 (train braking system failure), and M01 (poor safety education for workers). The hazard types included 1 M-, 5 H-, and 4 EM-type hazards. Although we obtained 20 H-type hazards from the accident reports, which only constituted 22% of all hazard nodes, the critical H-type hazards still constituted 50% of the total critical hazard nodes, which implies that human failure was the main cause of accidents. Based on the obtained critical hazard nodes and their parent nodes, hazard control measures can be implemented. For example, the top two causes of hazard event H02 were the conductor’s mistake (H03) and false signal displayed (EM04). For hazard node H03, we could not identify the causes from the accident reports. Therefore, railway safety managers should employ professional or experienced conductors to reduce these mistakes. Additionally, hazard node EM04 is a critical hazard node, which is mainly caused by EM03 (power supply failure). Therefore, the power supply departments of railway companies should strive to maintain a stable power supply.

The top 10 critical hazard nodes did not include E-type hazard nodes because the railway system is more robust to environmental changes than other transportation modes [39]. If the controllability of hazards is not analyzed, then E03 (wind) will constitute one of the top 10 critical hazard nodes. However, the wind cannot be alleviated because it is a natural hazard; as such, railway companies strive to protect railway systems from wind damage by technology improvement. In summary, the controllability of hazard factors must be analyzed to obtain feasible hazard control measures.

### 3.4. ACN Model Comparison

The proposed WLDACN model uses a different length metric method from other network-based accident causation models such as DACN and WDACN. To validate the efficiency of the WLDACN, we compared the performances of the five CHI methods under three accident causation models. The comparison results are listed in Table 3.

All three accident causation models performed the best when they were implemented using the HBA and IPM methods. However, the proposed IPM performed slightly better than the HBA method, although different accident causation models were used. Because WDACN and DACN obtain the edge length by reversing or disregarding the weights of edges, as described in Section 2, the proposed WLDACN performed the best among the three accident causation models. The WDACN and WLDACN demonstrated identical performance when implemented using the HDA and HPA methods because they used the same method to compute the weights of the edges. In general, combining the IPM and WLDACN yielded the best accident prevention results.

### 3.5. Limitation of the Method

The railway company may use facilities with various levels of reliability or employ staff with different professional abilities, which may lead to uncertain accidents. Therefore, the causality network obtained from different railway companies may be different. The obtained results only represent the dataset in this paper. Although the causality network based critical hazard identification method can help safety managers understand the causes of railway accidents to avoid similar occurrences in the future, the number of accident reports determines the efficiency of the method. In fact, the railway industry has developed for a long time, and we believe that the accumulated accident data are so large that the method can be used to suggest critical hazards.

## 4. Conclusions

Accident causation models enable us to effectively understand the causes of railway accidents. In previous ACN models, the effects of the frequency of hazard events or accidents are typically simplified; hence, the MPCP from hazards to accidents is inconsistent with the obtained SCP. Therefore, we proposed a new ACN model, where the MPCP problem is transformed into an SCP problem. The nodes in the network can be classified as hazard or accident nodes. As such, we can use the distance from hazards to accidents to measure the difficulty of various hazards causing accidents; consequently, critical hazards can be identified.

The causes of railway accidents should be understood to prevent accidents through hazard control. Therefore, we proposed CHI models to select critical hazards. The controllability of hazards, which is often disregarded, was discussed to determine the variables in CHI models. In this study, five CHI models were proposed to identify the critical hazard nodes in the ACN. To compare the performances of the models, three indices that can measure the efficiency of accident prevention were proposed. Comparison results from the case study indicated that the proposed IPM and WLDACN models performed better than the other models.

Because the results from the proposed model were based on real-world data, they offer useful insights into the hazard management of railway systems. The proposed model can suggest critical hazard factors that should be controlled. Safety managers can select hazard control options based on the critical causes of accidents.

In future studies, more accident data will be used to validate the proposed CHI model. In our model, the results of each type of accident, such as damage, injury, or death, were not considered due to inadequate data. Therefore, more data regarding accident damage should be obtained to measure the severity of accidents, and the weight of distance from hazards to each type of accident can be measured. Furthermore, the management cost of each hazard can be differentiated in future models.

## Figures and Tables

**Figure 1 entropy-23-00864-f001:**
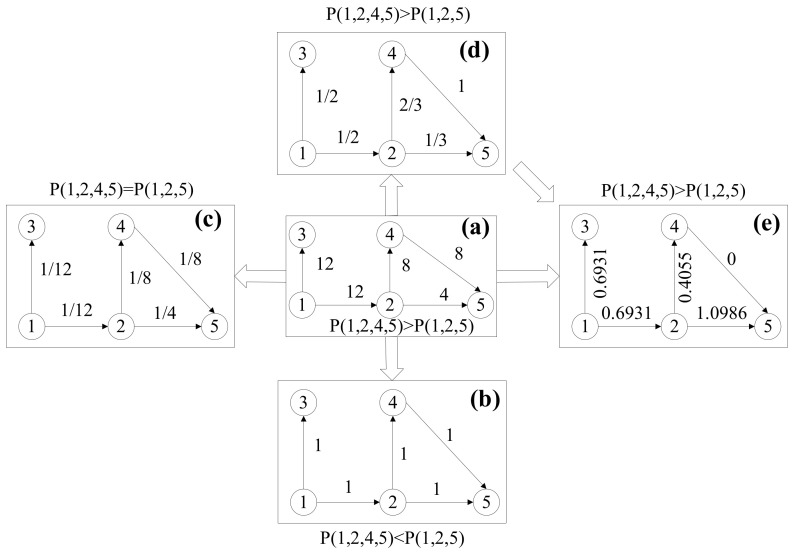
Accident causation network. (**a**) WDACN, (**b**) DACN (**c**) ACN after reversing the weight of the edge, (**d**) Normalized WDACN, (**e**) WLDACN.

**Figure 2 entropy-23-00864-f002:**
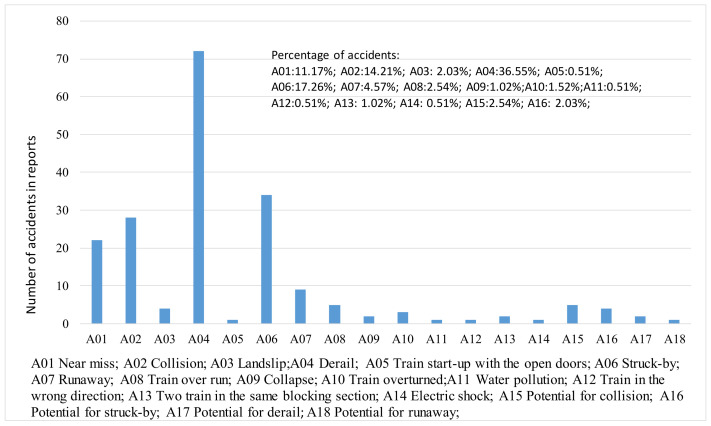
Collection of accidents from RAIB.

**Figure 3 entropy-23-00864-f003:**
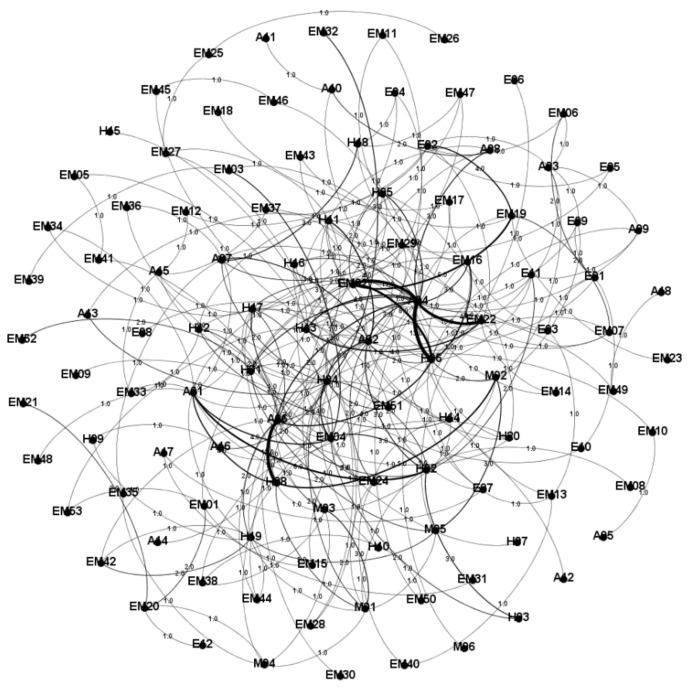
ACN with edge weight metrics.

**Figure 4 entropy-23-00864-f004:**
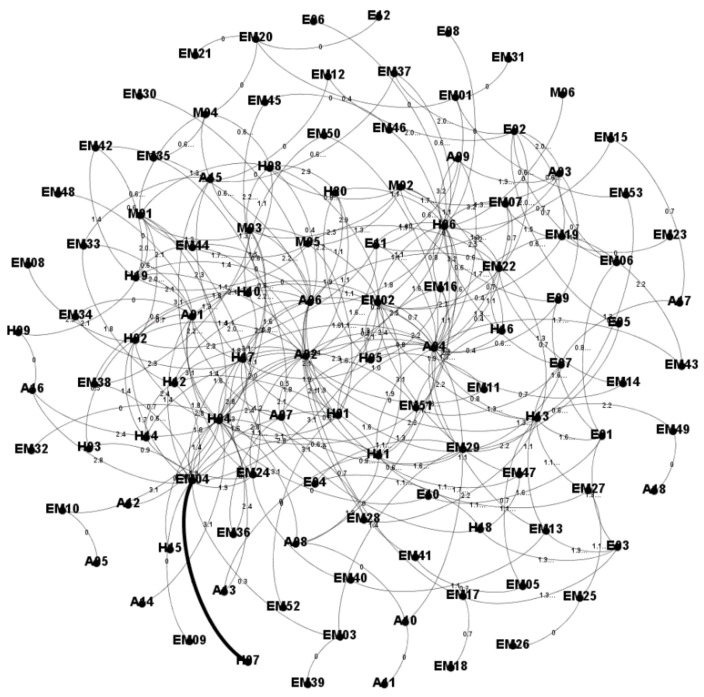
ACN with edge length metrics.

**Figure 5 entropy-23-00864-f005:**
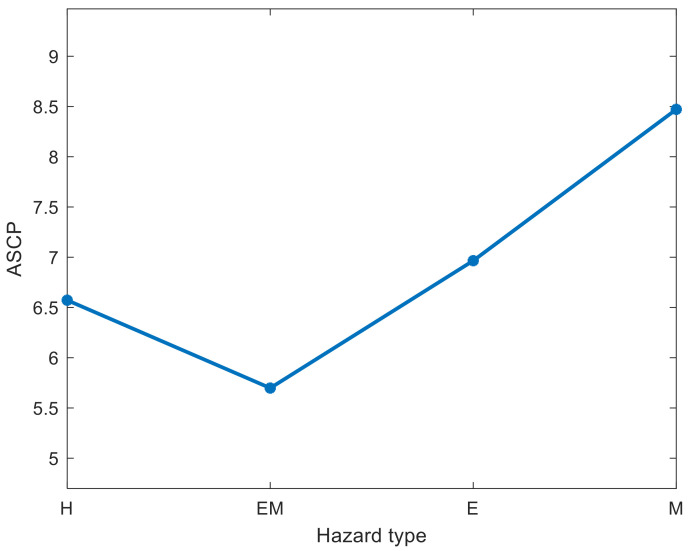
ASCP from each hazard type to accidents.

**Figure 6 entropy-23-00864-f006:**
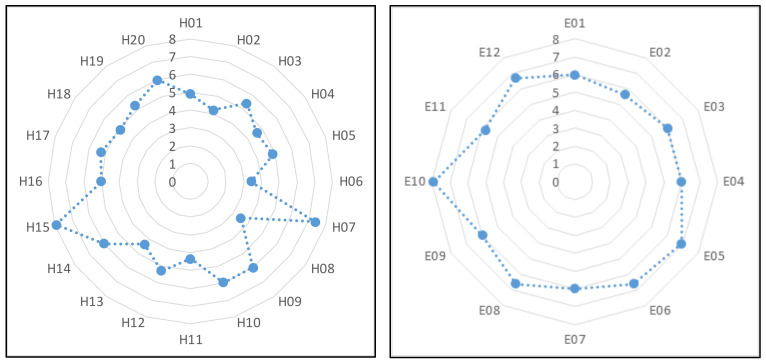
ASCP from each H-type hazard node and E-type hazard node to accidents.

**Figure 7 entropy-23-00864-f007:**
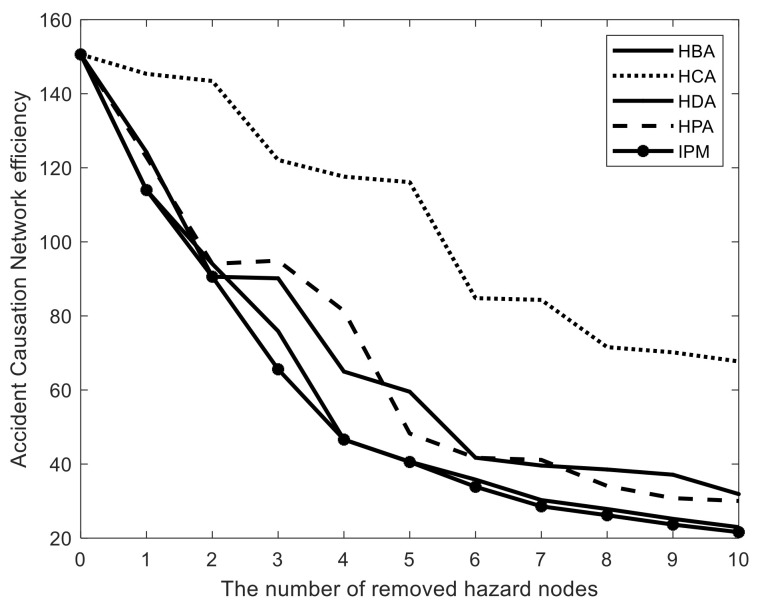
Comparison of different CHI methods.

**Figure 8 entropy-23-00864-f008:**
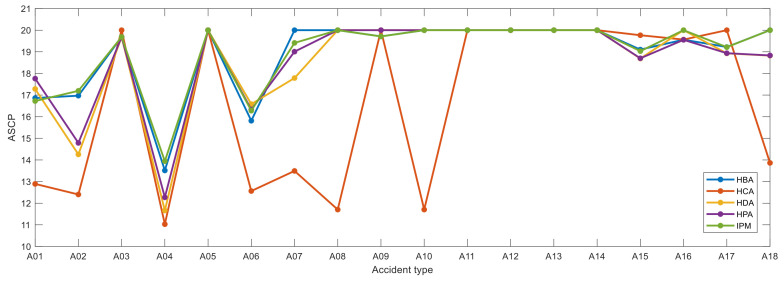
ASCP from all hazards to each accident type.

**Figure 9 entropy-23-00864-f009:**
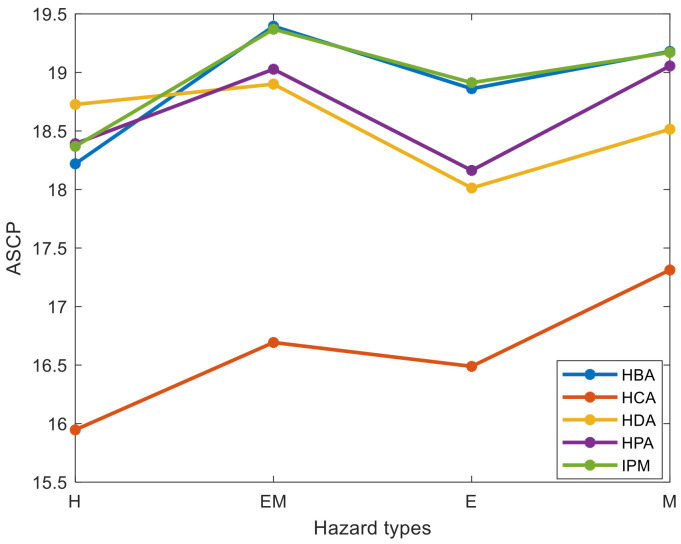
ASCP from each hazard type to accidents.

**Table 1 entropy-23-00864-t001:** Notations and abbreviations.

Variable	Description
Abbreviations	
CHI	critical hazard indentification
ACN	accident causality network
WDACN	weighted direct accident causality network
DACN	direct accident causality network
SCP	shortest causation path
MPCP	most probable causation path
WLDACN	directed ACN with weights and length metrics
HDA	high degree adaptive
HBA	high betweenness adaptive
HCA	high closeness adaptive
HPA	high pagerank adaptive
ASCP	average SCP
IPM	Integer Programming method
Notations	
h	hazard node
a	accident node
$w\left(i,j\right)$	the weight of the edge from node i to j
$F_{k}\left(h,a\right)$	The frequencies of the causation route k from hazard node h to accident node a
$S_{k}^{h,a}$	the set of points on causation route k
$p\left(i,j\right)$	the normalized weight of the edge from node i to j
$P_{k}\left(h,a\right)$	the active probability of causation route k
$l\left(i,j\right)$	the length of the edge from node i to j
E	the ACN efficiency
$D_{h}$	degree centrality of hazard node h
$B_{h}$	betweenness centrality of hazard node h
$C_{h}$	closeness centrality of hazard node h
$R_{h}$	PageRank of hazard node h

**Table 2 entropy-23-00864-t002:** Distances from 4 types of hazards to 18 types of accidents.

Accident Type	A01	A02	A03	A04	A05	A06	A07	A08	A09
H-type hazard	4.19	4.35	--	3.42	5.73	3.50	4.89	3.71	5.98
EM-type hazard	4.54	5.16	598	3.54	5.98	5.47	5.07	3.83	7.04
E-type hazard	5.80	5.98	5.98	5.62	9.03	6.90	5.98	5.91	7.37
M-type hazard	5.88	6.09	--	5.19	8.34	4.78	5.58	5.47	8.47
All types	5.10	5.39	5.98	4.44	7.27	5.16	5.38	4.73	7.22
Accident type	A10	A11	A12	A13	A14	A15	A16	A17	A18
H-type hazard	3.71	3.71	5.73	5.04	5.73	5.73	5.04	--	5.76
EM-type hazard	3.83	3.82	7.58	6.89	7.58	5.29	5.92	5.98	5.98
E-type hazard	5.91	5.91	9.03	8.34	9.03	5.98	8.34	5.98	8.06
M-type hazard	5.47	5.47	8.34	7.64	8.34	8.29	7.37	--	9.21
All types	4.73	4.73	7.67	6.98	7.67	6.32	6.67	5.98	7.25

**Table 3 entropy-23-00864-t003:** Difficulty of hazards causing accidents under different ACN models and CHI methods. (Bold represent the optimal values).

Models	HBA	HCA	HDA	HPA	IPM
DACN	27.3342	72.3560	32.4598	31.8879	27.1856
WDACN	25.8917	70.3325	31.8904	30.0392	25.3462
WLDACN	**22.9560**	**67.7097**	**31.8904**	**30.0392**	**21.6170**

## Data Availability

The data presented in this study are openly available in https://www.gov.uk/government/publications/raib-investigation-reports-and-bulletins (accessed date 6 July 2021).

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
