# Peer review of "Causality-Network-Based Critical Hazard Identification for Railway Accident Prevention: Complex Network-Based Model Development and Comparison"

_entropy, 2021, doi:10.3390/e23070864_

Round 1
Reviewer 1 Report
This work proposes a method to identify the biggest hazards in a causality network of hazards and accidents related to railway.
The novelty of this work is limited and the text is not that easy to read and needs improvement.
The abstract does not provide a clear description of the objective of the study as well as the contribution of the authors.
Too many abbreviations exist throughout the text that make it difficult to read (SCP, CHI, ACN, WDACN, WLDACN, HDA, HBA, HCA, HPA, DC, ASCP etc.).
Author Response
Reviewer1' comments
Point 1:This work proposes a method to identify the biggest hazards in a causality network of hazards and accidents related to railway. The novelty of this work is limited and the text is not that easy to read and needs improvement. The abstract does not provide a clear description of the objective of the study as well as the contribution of the authors.
Response 1:Thanks for your suggestion. We have revised the abstract and present the contributions clearly in section 1.3. The manuscript has been edited professionally. The certificate has been attached.
Point 2: Too many abbreviations exist throughout the text that make it difficult to read (SCP, CHI, ACN, WDACN, WLDACN, HDA, HBA, HCA, HPA, DC, ASCP etc.).
Response 2: Thanks for your suggestion. We have added table 1 to describe the notations and the abbreviations.

Reviewer 2 Report
The authors test alternative critical hazard identification (CHI) methods for railway accidents prevention, applying accident causation networks, using accident reports as the data. The manuscript was not the easiest and most clear to read, but somewhat understandable. I hope these comments and notes help the authors to improve their text.
The idea of identifying causalities from the written accident reports is innovative. However, the authors should describe more in detail, how this part of the work was conducted: manually by a human, or based on semantics analysis and programming? In case of the latter, closer description of the method is needed.
The design of the study is ok, but there are some issues with the interpretation of the results and their applicability throughout the article. Critical evaluation of the limitations arising from the data and the method are mostly missing, too.
The authors present the resulting critical hazards of the case study as if they were global critical hazards, although the result is based on a fully data-driven method and a national (United Kingdom) dataset. It is not brought out that the framing of the models is fully dependent on the data and thus the results only represent the datset in question. Before making any kinds of generalizations, the results should be validated. In this manuscript no validation is conducted at all, thus the results only reflect the data used and this should be brought out clearly.
In addition, throughout the article, the method should be communicated so that it only can suggest critical hazard factors for a closer analysis. A careful decision analysis is needed to be able to evaluate what might be the most effective or cost-effective combination of controll measures (given the available resources for the investments). It may be a better option to controll five less important hazard factors than the one ranked as the most critical, if the costs of these two strategies are somewhat the same. Importantly, the control actions seldom remove the possibility of the hazard or accident, but lower their probabilities with different levels of efficiency. On top of that, each action is associated with a certain level of implementation uncertainty. Considering these aspects, decision analysis is applied to find the most effective or cost-effective action to take, or the best controll strategy ( a series of actions), given the budget available.
More detailed comments:
The explanation of the figure 1 (Lines 146-163) is confusing and should be improved – it was really hard to follow. For example, the meaning of expressions like ”the frequency of the causation route” or ”active probability of the causation path” or ”maximum active probability” felt weird and their actual meaning should be explained. Also, check the sentence on lines 155-158: Is it related to the small figure (b) or (c)? What does ”a to e” refer to (line 156)? And why are the small figures (d) and (e) included, when they are not explained or referred to at all?
Figure 9: On the lines 392 - 393, it’s said ” As shown in Figure 9, the IPM and HBA methods performed better than the other three methods…”. However, this is not the case for the H-category, where HDA seems to perform best. Where does this phenomenon come from? Further on, why don’t you suggest using different methods for different hazard types? On the lines 412-413, it is stated that ”…the critical H-type hazards still constituted 50% of the total critical hazard nodes, implying that human failure was the main cause of accidents.” Thus I think my question is relevant, especially regarding the H-category.
Table 2: Explain the unit of the values presented (where do the numbers come from / how to interpret them?).
It is stated on lines 465 – 467, ”Furthermore, hazard control measures can be derived from the ACN because hazard managers can create hazard control plans based on the critical causes of accidents.” This is not quite true. ACN can help thinking of the potential control options, but the thinking should not be limited to the factors having nodes in the model only, as the framing of the model is fully based on the accident reports, that are based on the thinking of the reporting person(s). All in all, the quality and limitations of the data should be critically evaluated somewhere in the article.
Lines 472-473: ” The cost of hazard control is equal to the number of deleted hazard nodes in the proposed CHI model.” This statement is not true. The costs of divergent control options may be of totally different magnitude, thus one huge action may cost more than five smaller actions together.
Author Response
First we would like to thank reviewers and the editor for the positive and constructive comments and suggestions.
Reviewer 2 comments and responds
Point 1: The authors test alternative critical hazard identification (CHI) methods for railway accidents prevention, applying accident causation networks, using accident reports as the data. The manuscript was not the easiest and most clear to read, but somewhat understandable. I hope these comments and notes help the authors to improve their text.
Respond 1: Thanks for your suggestion. The manuscript has been edited professionally. The certificate has been attached.
Point 2: The idea of identifying causalities from the written accident reports is innovative. However, the authors should describe more in detail, how this part of the work was conducted: manually by a human, or based on semantics analysis and programming? In case of the latter, closer description of the method is needed.
Respond 2: Thanks for your suggestion. The causalities are extracted manually in order to ensure the accuracy in this paper. We did not use the semantics analysis and programming method because the method may miss some important information. Therefore, we will develop a well-performed machine learning method to extract causalities in the future studies
Point 3: The design of the study is ok, but there are some issues with the interpretation of the results and their applicability throughout the article. Critical evaluation of the limitations arising from the data and the method are mostly missing, too.
Respond 3: Thanks for your suggestion. The critical evaluation of the method limitations has been described in the section 3.5 of the revised manuscript. “The railway company may use facilities with various levels of reliability or employ staff with different professional abilities, which may lead to uncertain accidents. Therefore, the causality network obtained from different railway companies may be different. The obtained results only represent the dataset in this paper. Although the causality-network-based critical hazard identification method can help safety managers under-stand the causes of railway accidents to avoid similar occurrences in the future, the number of accident reports determines the efficiency of the method. In fact, the railway industry has developed for a long time, and we believe that the accumulated accident data are so large that the method can be used to suggest critical hazards.”.
Point 4: The authors present the resulting critical hazards of the case study as if they were global critical hazards, although the result is based on a fully data-driven method and a national (United Kingdom) dataset. It is not brought out that the framing of the models is fully dependent on the data and thus the results only represent the datset in question. Before making any kinds of generalizations, the results should be validated. In this manuscript no validation is conducted at all, thus the results only reflect the data used and this should be brought out clearly.
Respond4: Thanks for your suggestion. As you say, the obtained results only reflect the data used actually. For railway systems in other countries, we should establish the WLDACN based on the local accident reports. The manuscript focuses on the CHI method. Therefore, much more or big data should be used if we want to find the global critical hazards. In the future study, we will use the method to find the critical hazards in other railway systems such as USA, China and Australia. we believe that the accumulated accidents data is so big that the method can be used to suggest the critical hazards
Point 5: In addition, throughout the article, the method should be communicated so that it only can suggest critical hazard factors for a closer analysis. A careful decision analysis is needed to be able to evaluate what might be the most effective or cost-effective combination of control measures (given the available resources for the investments). It may be a better option to control five less important hazard factors than the one ranked as the most critical, if the costs of these two strategies are somewhat the same. Importantly, the control actions seldom remove the possibility of the hazard or accident, but lower their probabilities with different levels of efficiency. On top of that, each action is associated with a certain level of implementation uncertainty. Considering these aspects, decision analysis is applied to find the most effective or cost-effective action to take, or the best control strategy ( a series of actions), given the budget available.
Respond 5: Thanks for your suggestion. In the manuscript, we desire to find the critical hazards which contribute to the railway accidents greatly. However, it is hard to evaluate the cost of alleviating hazards and thus the heterogeneous control cost is seldomly considered. Therefore, we assume that the control cost increases as the number of alleviated hazards increases. In the proposed WLDACN, the network efficiency is used to reflect the difficulty of hazards causing accidents. The lower the network efficiency, the longer is the SCP from hazards to accidents and the accident prevention can be achieved. Therefore, the objective of CHI model is to minimize the network efficiency. Although the control cost is not differentiated, the obtained critical hazards can also help safety managers select control options. In the future studies, we hope to find a method to evaluate the control cost of each hazard and more data regarding accident damage can be obtained to measure the severity of accidents. We believe that the obtained methods and results can be improved after considering control cost and severity of accidents
More detailed comments:
Point 6: The explanation of the figure 1 (Lines 146-163) is confusing and should be improved – it was really hard to follow. For example, the meaning of expressions like ”the frequency of the causation route” or ”active probability of the causation path” or ”maximum active probability” felt weird and their actual meaning should be explained. Also, check the sentence on lines 155-158: Is it related to the small figure (b) or (c)? What does ”a to e” refer to (line 156)? And why are the small figures (d) and (e) included, when they are not explained or referred to at all?
Respond 6: Thanks for your suggestion. We have added the definition of the frequencies of the causation route in section 2.2.1. The causation routes are defined as the set of links from source hazard node to the accident node. There may be more than one route from source hazard node to the accident node and the same route may appear several times. The frequencies of the causation route from hazard node to accident node can be defined as , where denote the set of points on causation route . The active probability of the causation path has been expressed in Eq.(3) in the revised manuscript and it reflects the conditional probability of causation path given the occurrence of source hazard. In this study, the causation route with the highest active probability is defined as the most probable causation path. We have added table 1 to describe the notations and the abbreviations. We also have added the citation of figure1 (d) and (e) in the section 2.1.2 of the revised manuscript. We also add the derivation process of Eq.(5) which is a measure method of route uncertainty so that the manuscript fit better with the scope of entropy
Point 7: Figure 9: On the lines 392 - 393, it’s said ” As shown in Figure 9, the IPM and HBA methods performed better than the other three methods…”. However, this is not the case for the H-category, where HDA seems to perform best. Where does this phenomenon come from? Further on, why don’t you suggest using different methods for different hazard types? On the lines 412-413, it is stated that ”…the critical H-type hazards still constituted 50% of the total critical hazard nodes, implying that human failure was the main cause of accidents.” Thus I think my question is relevant, especially regarding the H-category.
Respond 7: Thanks for your suggestion. It is right that the ASCP of H-type hazard can be lengthened by HDA method to a higher level. However, the overall performance of HDA is worse than the IPM and HBA methods. Furthermore, the objective of CHI model is to lengthen the distance from all hazards to accidents instead of the distance from only human hazards to accidents. Therefore, the HDA performs worse in terms of the overall model efficiency. In the future study, we may use the proposed CHI model to investigate the human hazards only.
Point 8: Table 2: Explain the unit of the values presented (where do the numbers come from / how to interpret them?).
Respond 8: Thanks for your suggestion. There is no units of the values presented in Table 3 of the revised manuscript (Table2 in original manuscript). Table.3 present the network efficiency under different ACN models and CHI methods. Because the network efficiency reflects the difficulty of hazards causing accidents, therefore, table.3 present the accident prevention performance under different ACN models and CHI methods. In order to present the results accurately, we have revised the tile of table.3
Point 9: It is stated on lines 465 – 467, ”Furthermore, hazard control measures can be derived from the ACN because hazard managers can create hazard control plans based on the critical causes of accidents.” This is not quite true. ACN can help thinking of the potential control options, but the thinking should not be limited to the factors having nodes in the model only, as the framing of the model is fully based on the accident reports, that are based on the thinking of the reporting person(s). All in all, the quality and limitations of the data should be critically evaluated somewhere in the article.
Respond 9: Thanks for your suggestion. We have revised the sentence as “The safety managers can select hazard control options based on the critical causes of accidents.” The critical evaluation of the method limitations has been described in the section 3.5 of the revised manuscript.
Point 10: Lines 472-473: ” The cost of hazard control is equal to the number of deleted hazard nodes in the proposed CHI model.” This statement is not true. The costs of divergent control options may be of totally different magnitude, thus one huge action may cost more than five smaller actions together.
Respond 10: Thanks for your suggestion. The sentence has been deleted. In the manuscript, we assume that the control cost increases as the number of alleviated hazards increases and the heterogeneous control cost is seldomly considered. We believe that the obtained methods and results can be improved after considering control cost and severity of accidents in the future studies.

This manuscript is a resubmission of an earlier submission. The following is a list of the peer review reports and author responses from that submission.